# Community Perspectives on Intimate Partner Violence During Pregnancy: A Qualitative Study from Rural Ethiopia

**DOI:** 10.3390/ijerph22020197

**Published:** 2025-01-29

**Authors:** Zeleke Dutamo Agde, Jeanette H. Magnus, Nega Assefa, Muluemebet Abera Wordofa

**Affiliations:** 1Department of Population and Family Health, Institute of Health, Jimma University, Jimma P.O. Box 378, Ethiopia; mulu_abera.ts2009@yahoo.com; 2Department of Reproductive Health, College of Medicine and Health Sciences, Wachemo University, Hossana P.O. Box 667, Ethiopia; 3Faculty of Medicine, University of Oslo, 0316 Oslo, Norway; j.h.magnus@medisin.uio.no; 4College of Health and Medical Sciences, Haramaya University, Harar P.O. Box 138, Ethiopia; negaassefa@yahoo.com

**Keywords:** attitude towards marital violence, cultural norms about violence, community perspectives of violence, intimate partner violence, rural Ethiopia

## Abstract

Intimate partner violence (IPV) during pregnancy is closely associated with adverse maternal and fetal outcomes. To develop prevention strategies and interventions, the exploration of cultural norms, societal attitudes, and perceptions related to IPV is vital. This study explored community perspectives on IPV during pregnancy in rural Ethiopia. We used an exploratory qualitative study design to collect data. Data were collected through in-depth interviews (IDIs) and focus group discussions (FGDs) guided by a semi-structured topic guide. The data were analyzed using a thematic analysis approach, revealing the following four themes: (1) threats to the health of the mother and the fetus; (2) the contributing factors of IPV during pregnancy; (3) coping strategies for IPV during pregnancy; and (4) the need for intervention. Supportive attitudes toward IPV, early marriage, lack of awareness among offenders about its consequences, alcohol use, poor couple communication, and provocation by wives were found to be the causes of IPV during pregnancy. Participants in this study perceived IPV as a normal and unavoidable aspect of marital relationships. Comprehensive interventions that address challenging the cultural norms that condone IPV, increase community awareness of its detrimental effects, improve couples’ communication skills, and address alcohol abuse among men could play a crucial role in preventing or reducing IPV during pregnancy.

## 1. Introduction

Intimate partner violence refers to any behavior within an intimate relationship that causes physical, psychological, or sexual harm, including acts of physical aggression, sexual coercion, psychological abuse, and controlling behaviors, typically by a current or former intimate partner or spouse [1]. It is a major public health problem and a violation of women’s human rights [2]. The World Health Organization (WHO) report shows that globally, approximately 1 in 3 women experience physical and/or sexual violence from an intimate partner in their lifetime. According to this report, one of the most affected regions is Sub-Saharan Africa, with a prevalence rate of 33% [3]. A recent global systematic review and meta-analysis has revealed that globally, 18.7% of pregnant women experience some form of IPV, with physical violence (13.8%) being the most prevalent, followed by psychological (8.1%) and sexual violence (6.6%). Africa exhibits the highest rates, with 36.1% of women experiencing IPV during pregnancy [4].

The 2016 Ethiopia Demographic and Health Survey (EDHS) revealed that 34% of women aged 15–49 who had ever been married reported experiencing IPV. Of these, 24% experienced physical violence, 24% experienced psychological or emotional violence, and 10% experienced sexual violence [5]. Research has found that Ethiopia is one of the countries with the highest IPV during pregnancy, ranging from 26% to 65% [6,7,8,9].

Intimate partner violence during pregnancy has been associated with adverse maternal outcomes such as the risk of miscarriage, abortion, antepartum hemorrhage, hypertension, pre-mature labor, gestational diabetes, placental problems, infections, and mood disorders [10,11,12,13]. Furthermore, it is also associated with poor fetal outcomes such as preterm birth, intrauterine fetal death, small gestational age, and low birth weight [14,15].

Cultural norms, such as the traditional beliefs that men are superior to women and that a husband has the right to discipline his wife [16,17], are important factors in the prevalence of IPV. Beliefs that IPV is a private and family matter and societal attitudes, such as accepting IPV and the normalization and stigma surrounding reporting IPV, also play a pivotal role in its occurrence [18,19,20]. To our knowledge, there is a dearth of research, except for a few limited studies [21], on community attitudes and cultural norms regarding IPV during pregnancy, particularly in rural Ethiopia. Understanding community perspectives in a specific cultural context is critical, as members of the community significantly impact the experience of IPV during pregnancy and provide valuable insights for culturally sensitive interventions [22]. Thus, the aim of this study was to explore perspectives on IPV during pregnancy in an Ethiopian rural community.

## 2. Theoretical Framework

### The Socio-Ecological Theory

Socio-ecological theory was used as a framework with which to explore community perspectives on IPV during pregnancy [23]. As outlined by this theory, there are individual, relational, community, and societal factors that influence human behaviors [24]. The ecological perspective is an all-inclusive and multidimensional framework for understanding multiple factors influencing the experience of IPV against women [25]. The theory emphasizes that behavior is not solely determined by individual characteristics but is influenced by interactions among various levels of the social ecology. Understanding how these multiple level factors are related to violence is one of the important steps in the public health approach to preventing violence [21,26]. The choice of this framework is rooted in its ability to holistically capture the complex interplay of factors influencing IPV, particularly in rural settings where cultural norms and community structures are pivotal [16,24].

## 3. Methods and Materials

### 3.1. Study Design/Approach

An exploratory qualitative study design was used to collect community perspectives on IPV during pregnancy. In this study, data were gathered on community perspectives through focus group discussions (FGDs) and in-depth interviews (IDIs), guided by a semi-structured topic guide.

### 3.2. Study Area and Period

This study was conducted in Hadiya zone, which is one of the 11 zones in Central Ethiopia. Hossana is an administrative town in the zone and is located 235 km from Addis Ababa, the capital city of Ethiopia. The details about the study area are found in our previously published work [27]. The study took place between December 2022 and January 2023.

### 3.3. Study Participants

The study participants were purposively selected to take part in the research. The principal investigator collaborated with health extension workers (HEW) working in the area to identify participants. Health extension workers are government-employed individuals who work in each kebele, the lowest administrative unit in Ethiopia. To ensure diverse perspectives, male partners, religious leaders, community leaders, and the Women’s Development Army (WDAs) were selected for FGDs. Additionally, victims (pregnant women who experienced violence during pregnancy) were selected for IDIs. To encourage openness and honesty during the discussions and interviews, only one participant was selected from each household. Over a two-month period, a total of 8 FGDs with 70 participants and 7 IDIs with pregnant women were conducted (see Table 1).

### 3.4. Data Collection

Participants for FGDs were selected voluntarily and organized into the following four homogeneous groups based on their role in the community: male partners; religious leaders; community leaders; and WDAs. The rationale for engaging volunteer participants with different roles in the community was to gain targeted insights on community perspectives towards IPV during pregnancy [28]. Pregnant women who experienced IPV were purposively selected for IDIs to gain a comprehensive understanding of their perspectives on IPV. The principal investigator conducted FGDs of male partners, religious leaders, and community leaders, while a female expert at conducting qualitative studies conducted the FGDs of WDAs and all IDIs. All FGDs and IDIs were conducted in a safe and private place.

Semi-circular seating was arranged to enable everyone to see, listen to, and engage with one another during the discussions. Prior to the start of the interviews and discussions, the research team introduced themselves and explained that they were conducting a study on women’s health. Participants were then given a brief explanation of the study’s purpose and assured that the information they shared would be kept confidential, as stated in the information sheet. The ground rules and regulations of the FGD were noted and made clear to the participants prior to the session [28].

A semi-structured topic guide was developed by reviewing various similar studies [19,24,29]. The guide focused on topics such as awareness and perceptions of IPV during pregnancy, causes of IPV during pregnancy, and coping and prevention strategies for IPV during pregnancy. During the course of discussions or interviews, the main questions were accompanied by a series of follow-up questions and probes aimed at gaining a more profound understanding of perspectives on IPV during pregnancy [28].

The topic guide was pretested with three pregnant women to identify any ambiguities and inconsistencies in the guide. Then, slight amendments were made after the pretest. The FGDs and IDIs were conducted using the local language (Hadiyyisa). All FGDs and IDIs were audio-recorded after obtaining permission from the participants. The FGDs in the male partner, religious leader, and community leader groups were moderated by the principal investigator, while the WDA groups were moderated by a female moderator who had previous experience in moderating FGDs. A research assistant observed and took notes on the non-verbal aspects of the situation throughout the group discussions. An experienced female interviewer conducted the IDIs. Each IDI or FGD typically lasted between 80 and 90 min. We felt that we had reached data saturation, with no new data emerging, after conducting 7 IDIs and 8 FGDs.

### 3.5. Data Processing and Analysis

Focus group discussions and IDIs were transcribed verbatim from audio recordings, along with field notes, into Hadiyisa (the local language). Then, they were translated into English to facilitate the analysis process. The transcripts were thoroughly reviewed, line by line, in order to gain a deep understanding of the content. The complete dataset was imported into ATLAS.ti 7.1 for data analysis. Data were analyzed using Braun and Clarke’s (2006) thematic analysis framework, employing hybrid (deductive and inductive) approaches [30]. To ensure reliability and minimize potential coding biases, two independent coders coded the data [31]. After a discussion, the codes were merged to establish consensus codes. These codes were then grouped into broader categories and overarching themes. The themes were carefully reviewed to capture the content and meaning of the data. The results were summarized and reported, including important direct quotes.

### 3.6. Trustworthiness

The credibility of the study was ensured by addressing key issues. The study included a concise explanation of the methods used for FGDs and IDIs, which encompassed participant selection, data analysis, and reporting procedures. Transcripts were thoroughly reviewed, line by line, to so that the researchers could become acquainted with the data and so that the data could be cross-checked with field notes. Additionally, the research team that participated in data collection had previous experience in conducting qualitative research. The research team, who were fluent in the local language (Hadiyyisa), conducted all FGDs and IDIs over a period of two months. This ensured consistency throughout the data collection process. In order to improve the study’s dependability, we included a comprehensive description of the research context, participants, data collection, and analysis procedures.

The findings were triangulated, and the transferability was enhanced by using multiple data sources (participants) and a mixed method of data collection (FGDs and IDIs) [32]. The findings were reported following the guidelines outlined in the consolidated criteria for reporting qualitative research (COREQ), which is a 32-item checklist [33].

### 3.7. Ethical Considerations

The study was approved by the Ethics Committee of Jimma University’s Institutional Review Board on 8 November 2022 (JUIH/IRB/222/2022). Permission was obtained from local officials before the preceding data collection. The participants were informed of the purpose and procedure of the research and their right to not participate or withdraw from the study at any time. We obtained informed written consent from all study participants individually. Each participant was assigned a unique anonymous code to maintain confidentiality. Study participants were assured that all study information would be securely kept and treated as confidential. Additionally, they were instructed not to mention their names or the names of other participants during the discussion.

## 4. Results

Four themes and 22 categories emerged from all FGDs and IDIs on community’s perspectives on IPV during pregnancy. The themes include, as follows: threats to the health of the pregnant women and developing fetus; contributing factors of IPV during pregnancy; the coping strategies of IPV during pregnancy; and the need for intervention (see Table 2).

### 4.1. Threats to the Health of the Pregnant Woman and the Developing Fetus

#### 4.1.1. Experience of IPV During Pregnancy

This study revealed that experiencing IPV during pregnancy was common among pregnant women. It took the form of physical violence, including beatings or hitting, that resulted in injuries including cuts, bruises, aches, deep wounds, and life-threatening conditions. Pregnant women also experienced psychological IPV such as verbal abuse, withholding affection, limiting social interactions, insults, threats to harm, intimidation, and constant shouting. Sexual violence, specifically sexual intercourse without the wife’s consent, was rarely discussed as a form of IPV during pregnancy.

#### 4.1.2. Expectations of Gender Roles

Participants discussed that pregnant women were expected by male partners and family members to carry out all household activities during pregnancy as if they were not pregnant. This workload during pregnancy is perceived as a major challenge. Not fulfilling household activities leads the couple into conflict and, ultimately, violence:


*…A wife has to complete all her household responsibilities on time. If she doesn’t, it frequently results in disputes and, in some cases, more serious problems. It is her responsibility to manage the household, whether she is pregnant or not.*
(FGD, male partner)

Discussants in this study reported that men exercised control and dominance in their homes, which frequently resulted in violence when their authority was challenged. This power and control dynamic created an environment in which any perceived danger to male dominance, whether through questioning or opposition, elicited aggressive and violent responses:


*Wives should know that husbands are the powerhouses of the household. Wives should be under the control of us [husbands] and should accept whatever we, as husbands, say. Intimate partner violence during pregnancy often occurs when wives start to control their husbands, which needs to be corrected. Our cultural story tells of men having control over women, not vice versa.*
(FGD, male partner)

Participants expressed mixed arguments regarding the acceptability of IPV during pregnancy. Participants in the FGDs and IDIs mentioned that IPV is often seen as normal and inevitable, and is, notably, considered acceptable under certain circumstances such as when the wife has done something wrong, argues with the husband, or neglects the children. However, some participants strongly opposed this view, emphasizing the need for respect and the protection of pregnant women.


*…Many women believe that if their husbands beat them, it’s because they must have done something wrong. Wife beating is a normal and inevitable part of married life.*
(FGD, male partner)


*To keep the house in order, a husband must sometimes punish his wife. It is a part of our culture, and many of us [women] consider it necessary.*
(FGD, pregnant woman)

The other participants held the opposite attitude toward IPV:


*Beating a pregnant wife is never justified. It is not only harmful to the woman, but also to the unborn child. A husband should protect and support his wife, especially during pregnancy, rather than engage in any form of violence.*
(FGD, male partner)

### 4.2. Justification of IPV

In this study, participants perceived husbands as justified in hitting or beating their wives under certain conditions, such as arguing with their husbands, not carrying out household activities, questioning the husband’s authority, going out without permission, and not caring for children:


*If my wife argues with me, she deserves to be disciplined. It’s my responsibility to keep order in the house. I feel that hitting my wife is the only way to show her that she must respect my authority, especially if she goes out without my permission or neglects the children.*
(FGD, male partner)

However, contradictory views were observed among other FGD participants regarding men’s justifications for beating their wives:


*Regardless of the circumstances, violence toward one’s wife is never acceptable. Our faith instructs us to treat our wives with dignity and compassion.*
(FGD, religious leader)


*…Beating a woman is never justified in any situation. We must promote nonviolent dispute resolution and preserve the dignity of all individuals.*
(FGD, male partner)

### 4.3. Stress Related to Sons Preference

The participants reported that pregnant women face psychological stress as a result of pressure and mocking from their spouses and family members to have a male child. In society, preferences for male offspring are deeply ingrained regardless of economic status or educational status, whether educated or not. Participants felt that son preference is culturally and patriarchally ingrained and difficult to alter. Pregnant women mentioned having a male child to minimize discrimination and stigma by their husbands and family members. One participant shared:


*…The constant pressure to have a boy is overwhelming. My husband and his family are continuously assuring me that having a son will bring honor and stability to our family. It feels as if my worth is determined by my ability to bear a male child. I put my trust in God, expecting He will reward me with a son.*
(IDI, IPV victim)

### 4.4. Poor Social Support

Participants reported that a lack of emotional, psychological, and practical support from family and the wider community could create an environment where stress and frustration build up, often leading to increased conflict and violence


*When a pregnant woman doesn’t receive support from her family and community, it increases the tension at home. This lack of support can lead to more arguments and, unfortunately, more violence.*
(FGD, pregnant woman)

### 4.5. Absence of Legal Protection

Participants stated that, despite the WDA’s and community elders’ small efforts to address concerns and conflicts between husbands and wives, there is a lack of an adequate legal structure to protect pregnant women from IPV. The lack of legal safeguards permits abusers to continue abusing their wives without the fear of repercussions:


*Due to the lack of legal measures, perpetrators are able to abuse their wives. We have not witnessed any legal body taking measures against violence perpetrators.*
(FGD, WDA)

### 4.6. Contributing Factors of IPV During Pregnancy

#### 4.6.1. Men’s Alcohol Consumption

The participants in this study perceived that men’s use of alcohol was one of the main causes of IPV during pregnancy. Alcohol consumption by men was found to enhance tension and anger, leading to a greater incidence of IPV against pregnant women. Men’s heavy alcohol consumption was considered a source of conflict, impairing judgment, and aggravating violent tendencies in the home:


*When my husband drinks alcohol, he becomes more aggressive. The smallest disagreement can turn into a violent outburst (IDI, IPV victim). We see many cases where men who heavily drink alcohol and come to their home shout at their wives seeking the reason for argue as they feel powerful. Then, the husband end up with the violence!*
(FGD, community leader)

#### 4.6.2. Lack of Effective Communication

Participants reported that a lack of effective and sympathetic communication, particularly the absence of smooth and active listening, was perceived as one of the causes of IPV during pregnancy in marital relationships. The failure of partners to engage in open, polite communication and truly listen to each other’s problems usually resulted in misunderstandings, annoyance, and increased confrontations between them, in which emotions may swiftly escalate into violence, as grievances and issues were not addressed or handled in an acceptable manner.

#### 4.6.3. Lack of Awareness on Consequences of IPV

Participants reported that a lack of awareness about the consequences of violence during pregnancy contributed to the perpetuation of abusive behaviors in marital relationships. Many perpetrators did not recognize the serious health risks that such violence posed to both the women and their developing fetuses:


*We often beat, slap, or insult our wives without realizing the harm it causes. Many of us believe that such violence does not affect the health of the women or the developing fetus in the womb.*
(FGD, Male partner)

### 4.7. Provocation by Wife

Participants indicated that certain behaviors exhibited by pregnant women provoked their partners, thus leading to incidents of violence. These perceived provocations encompassed a variety of actions and situations, ranging from verbal or behavioral incitements to challenges to the partner’s authority:


*I beat my wife because she provoked me by saying, “Goonchikilas hino appise-beat me if you are a man”, which really hurt my heart. It was a direct assault on my authority and masculinity. My emotions overwhelmed me in that intense moment and I behaved without considering the repercussions.*
(FGD, male partner)

### 4.8. Early Marriage

Community members in this study reported that early marriage, particularly for girls under the age of 18, was one of the causes of IPV during pregnancy. Marrying at an early age frequently resulted in girls having a limited education, economic dependency, being less skilled at cooking food and performing home chores, and having less power over the relationship. The immaturity and lack of skills related to marital responsibilities at the household level increased the risk of IPV:


*I got married when I was 16. I didn’t know how to manage a household or stand up for myself. This made my husband think he could control everything, and if I made a mistake, he would get violent.*
(IDI, IPV victim)

### 4.9. Living with Extended Family

Participants mentioned that living with extended family members, such as in-laws (mother-in-law, father-in-law, sister-in-law, and brother-in-law), could be a source of conflict. The additional familial pressures and dynamics created an environment where tensions could escalate, resulting in increased instances of IPV. Financial reliance caused an imbalance of power and authority in the household, which contributed to the perpetuation of violence:


*Living with my in-laws adds so much pressure. My husband gets influenced by them, and if I can’t meet their expectations, he becomes violent. The financial dependence just makes things worse because I have no control over anything.*
(FGD, pregnant woman)

### 4.10. Economic Dependence

The study participants mentioned that the economic dependence of wives was perceived as a cause of IPV during pregnancy. Economic dependence often left women vulnerable and unable to challenge or leave abusive relationships, as they lacked the financial means to support themselves independently. The financial reliance caused an imbalance of power and authority in the household, which contributed to the perpetuation of violence:


*When a wife is completely dependent on her husband for financial support, he gains more power over her and can mistreat her more easily. She can’t confront him or leave because she has nowhere else to go.*
(FGD, WDA)

### 4.11. Coping Strategies for IPV During Pregnancy

#### 4.11.1. Concealment

Participants in this study stated that victims of IPV during pregnancy avoid exposing their experiences to third parties or seeking help outside of their house due to a strong fear of revenge from their abusers:


*Victims [pregnant women who have experienced IPV] are often afraid of retaliation from their husbands if they disclose the abuse. They fear that reporting the violence could lead to more severe abuse or even life-threatening consequences. We hear from somebody else. This is the real challenge in handling issues related to IPV.*
(FGD, WDA)


*…Because of the fear of judgment, shame, and embarrassment surrounding me, I kept silent. I normalized IPV instead of sharing it with someone, convincing myself that disclosing would only escalate the violence further.*
(IDI, IPV victim)

Participants mentioned that economic dependence on their abuser could make women reluctant to disclose the violence:


*They [wives] may fear losing financial support, housing, and stability if they leave or report the abuse. The fear of being unable to support themselves and their children without the abuser’s financial assistance can be a significant barrier.*
(FGD, community leader)

#### 4.11.2. Passive Behavior

Participants in FGDs and IDIs reported that IPV victims use passive conduct as a coping strategy during pregnancy. This included avoiding confrontations and attempting to create a tranquil family atmosphere to reduce the likelihood of subsequent violence. Victims choose to keep silent, obey their abusers’ demands, and endure the abuse silently, all in an effort to protect themselves and their unborn child from harm, while hoping that their husbands’ abusive behavior will subside in the future.

#### 4.11.3. Seeking Help

The study revealed that the surrounding familial and societal stigma of IPV made it difficult to seek help. Victims did not seek help because they felt ashamed and isolated. Victim-blaming attitudes, in which victims are held responsible for the violence against them, could further perpetuate a cycle of abuse and silence:


*I was ashamed to share the abuse because others would blame me and claim it was my fault. This made it difficult to seek assistance, and many of us suffer in silence.*
(IDI, IPV victim)

#### 4.11.4. Reconciliation

The study demonstrated that reconciliation was a culturally endorsed approach for IPV victims coping with IPV during pregnancy. Participants encouraged the victims’ reconciliation, as divorce or separation—a potential avenue for escape—was not permitted due to religious and societal stigmatization.

### 4.12. Need for Intervention

#### 4.12.1. Engagement of Men

Male partners in the FGDs recommended that engaging men in awareness-creating interventions, including educating them about the consequences of IPV during pregnancy and promoting healthy marital relationships, is crucial for reducing the incidence of IPV in the community. They emphasized that such initiatives would help men to understand the negative impacts of violence on both pregnant women and the developing fetus:


*Educating men about the harmful effects of IPV during pregnancy and how to build healthy relationships is essential. When men understand the consequences of their actions, it can significantly reduce violence in our community.*
(FGD, male partner)

#### 4.12.2. Engagement of Couple

Educating couples was one of the strategies recommended by participants in the FGDs to address incidences of IPV during pregnancy. They stressed the need to equip both partners with knowledge and skills in order to create healthy, respectful relationships and realize the detrimental implications of IPV:


*Educating couples about healthy communication and the harmful effects of violence can make a big difference. When both partners understand how to support each other, it helps to reduce IPV during pregnancy.*
(FGD, pregnant woman)

#### 4.12.3. Role of Religious Leaders

Religious leaders in the FGDs recommended actively engaging in supportive roles to assist and counsel the victims of IPV, seeing these roles as crucial in the community. They highlighted the importance of leveraging their positions of influence within the community to provide guidance, offer emotional and spiritual support, and promote non-violent, respectful relationships:


*…All religious leaders must take their own share in changing the attitudes of perpetrators of violence. We can help couples in our community develop more respectful and non-violent relationships by providing emotional and spiritual support.*
(FGD, religious leader)

#### 4.12.4. Role of Community Leaders

Community leaders discussed the necessity of their active participation in resolving marital disputes as one of the vital techniques for addressing IPV during pregnancy. They mentioned that their involvement in mediation and dispute resolution might help to reduce tensions and prevent violence against women.

## 5. Discussion

In this study, participants perceived IPV during pregnancy as a threat to pregnant women and their growing fetus, as well as having social and economic consequences. This perception highlights the critical need for programs that prevent IPV during pregnancy and promote the health and well-being of mothers and their unborn children. In line with Heise’s ecological model, the results demonstrate connections among individual, relationship, community, and societal factors that impact the understanding of IPV [34]. Therefore, this model demonstrates a comprehensive framework with which to illustrate the community perspectives on IPV during pregnancy in our study setting.

### 5.1. Individual Level

Women are especially vulnerable to IPV during pregnancy because of their economic reliance on their spouses and lack of autonomy in decision making. This dependence creates a power imbalance, increasing their susceptibility to abuse. Participants stated that victims of IPV during pregnancy are generally hesitant to leave abusive relationships, owing to financial restrictions and a fear of losing support for themselves and their unborn children. This emphasizes the vital need for economic empowerment and support structures for pregnant women experiencing IPV, which is in line with the findings of a systematic review of IPV in low- and middle-income countries by Vyas S. and Watts C., 2008 [35].

As in other studies in Ethiopia [36,37] and other countries [38,39,40], our study revealed that early marriage (below 18 years) is one of the causes of the IPV during pregnancy. This could be due to the fact that women who are married early might have limited education, be economically dependent on their husbands, and have less skill for marital responsibilities, which places them in a cycle of violence [35]. This finding underscores the importance of addressing early marriage as a critical factor in mitigating IPV during pregnancy.

In our study, several barriers, including fear of retaliation, stigma, isolation, and victim-blaming, considerably deterred victims from disclosing their experiences of IPV during pregnancy. These societal pressures create an environment where victims feel unsafe and unsupported in seeking help, leading them to adopt coping strategies such as concealment, passive behavior, and reconciliation. The economic dependence on their husbands further exacerbates this situation, as many victims fear the social and financial insecurity of leaving abusive relationships [21]. Consequently, these coping strategies become the only viable options for many women facing IPV during pregnancy. Similar findings have been noted in Ethiopia [21] and Nepal [19].

### 5.2. Relational Level

Participants in our study perceived that a lack of good communication between couples was one of the contributing factors to IPV during pregnancy. Poor husband–wife communication can lead to misunderstandings, unmet expectations, and unresolved arguments, exacerbating interpersonal tensions and, eventually, IPV [41]. This finding emphasizes the need for interventions focused on improving communication skills among couples, especially during pregnancy, to combat IPV in pregnancy.

Participants in our study mentioned that living with extended family members, such as in-laws, could be a source of conflict and ultimately result in IPV during pregnancy. This finding is supported by a study conducted in Kenya and Burkina Faso [42], which revealed that living with extended family members increases household stress and the likelihood of encountering IPV. Similar to other study findings [43,44], in our study, a husband’s alcohol consumption is directly related to IPV during pregnancy. Raising awareness about the negative effects of alcohol consumption on IPV can help to change the attitudes and behaviors of perpetrators [45].

The results of this study suggest that men’s lack of knowledge regarding the severe consequences of IPV during pregnancy contributes significantly to its occurrence, indicating that educating men about the detrimental effects of IPV not only on their partners but also on their unborn children could play a crucial role in reducing instances of such violence [46].

### 5.3. Community Level

As in other studies [47,48], son preference was reported as a significant source of conflict that leads to violence during pregnancy. Son preference is indeed one of the manifestations of subordination in patriarchal societies [49]. This preference reinforces gender inequalities and perpetuates violence against women, as it places a higher value on male offspring than female children. Interestingly, community members in our study mentioned that treating sons and daughters equally is an important strategy to address IPV.

Victims/survivors of IPV often seek help from informal sources, including family, friends, and neighbors [50,51]. Participants in our study expressed that the lack of emotional, psychological, and practical support for the victims could create an environment where stress, frustration, and cycles of IPV occur. Similar observations have been shown in other studies [6,52], where women with low social support are more likely to experience IPV and less likely to disclose IPV and seek help. This could be because women with less social support are less likely to break free from abusive relationships and seek alternative dispute resolution methods, as they are more likely to be constrained by social and traditional norms that urge women to stay in abusive relationships. This implies that community programs, support groups, and policies aimed at enhancing social connections and support for women, particularly those at risk of IPV, could play a significant role in prevention and recovery.

### 5.4. Societal Level

Research indicates that cultural norms and societal acceptance play a critical role in the prevalence of IPV [7,18,53]. Participants revealed that IPV during pregnancy was perceived by the participants as normal, inevitable, and acceptable under certain conditions. The societal environment and cultural norms facilitate and justify IPV that translates into individual experience. Participants justified IPV as a way to discipline a wife. Considerably, these perceptions contribute to the higher magnitude of IPV in societies where such behavior is regarded as normal [54]. This finding underscores the importance of addressing such beliefs in society in order to address IPV during pregnancy and improve maternal and fetal health outcomes.

The absence of legal protections for victims of IPV enables abusers to continue their harmful behaviors without fear of legal repercussions. In fact, in societies where legal frameworks are weak or poorly enforced, IPV incidents tend to increase [49]. This could be because perpetrators are aware that their actions are unlikely to result in legal consequences. To effectively prevent violence against pregnant women, it is important to involve many stakeholders and use various approaches due to the complex causes that contribute to violence.

### 5.5. Strength and Limitation of the Study

The study explored the various viewpoints of the community, which allowed the researchers to deeply explore the perspectives of IPV during pregnancy. The use of mixed methods of data collection (FGD and IDI) enabled a triangulation of the findings. The limitation of this study was doubt regarding whether the participants in FGDs or IDIs honestly expressed their views or simply shared what they believed to be socially acceptable or desirable to the researcher. However, all FGDs were facilitated by a gender-matched moderator, and the IDIs were conducted by an interviewer of the same gender.

## 6. Conclusions

The results of this study indicated that IPV during pregnancy is common and viewed as a normal and unavoidable part of marital partnerships. Several contributing factors of IPV during pregnancy were explored, including, as follows: favorable attitudes towards IPV; early marriage; the offenders’ lack of awareness of the repercussions of IPV and alcohol intake; poor communication between couples; and provocation by the wife. To effectively prevent or control IPV during pregnancy, comprehensive interventions that address cultural attitudes against IPV, raise awareness and educate about the detrimental effects of IPV, improve couples’ communication skills, and address men’s alcohol intake are crucial.

## Figures and Tables

**Table 1 ijerph-22-00197-t001:** Characteristics of study participants.

Participants	Age Range	Average Age	Number of Participants	Number of Focus Group Discussions (FGD)
Male partners	18–55	32	21	2
Community leaders	35–65	43	18	2
Religious leaders	25–65	38	15	2
Women’s development army	25–55	36	16	2
Pregnant women	15–45	29	7	NA
Total participants			77	8

NA: Not Applicable.

**Table 2 ijerph-22-00197-t002:** Themes and Sub-themes.

Themes	Sub-Themes
Threats to the health of the pregnant women and the developing fetus	Experience of IPV
	Expectations of gender roles
	Acceptance of IPV
	Justification of IPV
	Stress related to sons preference
	Absence of legal protection
	Poor social support
Contributing factors of IPV during pregnancy	Men’s substance abuse
	Lack of awareness on consequences of IPV
	Lack of effective communication
	Provocation by wife Living with extended family
	Early marriage
	Economic dependence
Coping strategies for IPV during pregnancy	Concealment
	Passive behavior
	Help seeking Reconciliation
The need for intervention	Engagement of men
	Engagement of couples
	Role of religious leaders
	Role of community leaders

## Data Availability

Data used for analysis in this study will be available upon reasonable request from the corresponding author.

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
