# Peer review of "Community Perspectives on Intimate Partner Violence During Pregnancy: A Qualitative Study from Rural Ethiopia"

_ijerph, 2025, doi:10.3390/ijerph22020197_

Round 1
Reviewer 1 Report
Comments and Suggestions for Authors
I found the article well-written methodologically as well as in the presentation of its findings and conclusions. This is how a qualitative study is presented. The choice of theoretical framework, too, was appropriate. The only flaw was the brevity of the section.
The theoretical framework is given very briefly. It should be presented in detail, maybe with a diagram for greater comprehension, and the reason for its choice explained.
The main question addressed by the research is how the social environment and cultural norms facilitate and justify IPV that translates into individual experience.
I do think that the topic is original and relevant to the field because IPV studies are done from the point of view of the victim. The impact of society and culture is gleaned from the narratives of the victims. This study turns it on its head and asks the community members and (you could say) perpetrators why IPV occurs. This is a new orientation whose findings can be compared to victim narratives for a further indepth understanding.
It adds perspectives of community and men to the subject area compared with other published material.
The Discussion presents in detail the perceptions of the community members and male partners at all the levels enunciated by the theoretical framework - individual, relational, community, societal. This section addresses the main question posed and the main findings are summarized in the conclusion.
The reference list has listed both global studies conducted by WHO as well as a large number of studies in Ethiopia or other African nations that are culturally similar. The references on qualitative research methods and thematic analysis are good too.
Table 1 is merely sample description and table 2 lists the themes and sub-themes. Nothing further can be added to them.
In sum, I found this to be a well-designed study and well-written manuscript.
Author Response
For review article: Community Perspectives on Intimate Partner Violence During Pregnancy: A Qualitative Study from Rural Ethiopia
Dear reviewer,
Dear reviewer, we would like to thank for your investing time and efforts made to provide the valuable and very crucial feedback to improve our manuscript. We would like to acknowledge that all the comments and questions raised are really important to further improve our manuscript. We have carefully addressed all the comments and questions raised point by point. We highlighted with a red color in the revised manuscript.
Response to Reviewer #1
I found the article well-written methodologically as well as in the presentation of its findings and conclusions. This is how a qualitative study is presented. The choice of theoretical framework, too, was appropriate. The only flaw was the brevity of the section.
Response: Thank you for your appreciation.
Comment 1: The theoretical framework is given very briefly. It should be presented in detail, maybe with a diagram for greater comprehension, and the reason for its choice explained.
Response: Thank you for pointing out this. We agree with this comment. Therefore, we have presented the detail of the theoretical framework and the reason for its choice is also explained in the revised manuscript under theoretical framework section as follows:
The ecological perspective is an all-inclusive and multidimensional framework for understanding multiple factors influencing the experience of IPV against women [25]. The theory emphasizes that behavior is not solely determined by individual characteristics but is influenced by interactions among various levels of the social ecology. Understanding how these multiple level factors are related to violence is one of the important steps in the public health approach to preventing violence [21, 26]. The choice of this framework is rooted in its ability to holistically capture the complex interplay of factors influencing IPV, particularly in rural settings where cultural norms and community structures are pivotal [16, 24].
Comment 2: The main question addressed by the research is how the social environment and cultural norms facilitate and justify IPV that translates into individual experience.
Response: Thank you for your insightful feedback. In response to your feedback, we have expanded the discussion section that address the societal environment and cultural norms facilitate and justify IPV that translates into individual experience. We have addressed in the revised manuscript as follows under discussion section as follows:
The societal environment and cultural norms facilitate and justify IPV that translates into individual experience.
Comment 3: do think that the topic is original and relevant to the field because IPV studies are done from the point of view of the victim. The impact of society and culture is gleaned from the narratives of the victims. This study turns it on its head and asks the community members and (you could say) perpetrators why IPV occurs. This is a new orientation whose findings can be compared to victim narratives for a further in-depth understanding.
Response: Thank you for recognizing the originality and relevance of our study.
Comment 4: It adds perspectives of community and men to the subject area compared with other published material.
Response: Thank you for highlighting the contribution of our study in adding the perspectives of community members and men to the subject area.
Comment 5: The Discussion presents in detail the perceptions of the community members and male partners at all the levels enunciated by the theoretical framework - individual, relational, community, societal. This section addresses the main question posed and the main findings are summarized in the conclusion.
Response: Thank you for highlighting the discussion presents in detail the perceptions of the community members and male partners at all the levels supported by theoretical framework.
Comment 6: The reference list has listed both global studies conducted by WHO as well as a large number of studies in Ethiopia or other African nations that are culturally similar. The references on qualitative research methods and thematic analysis are good too.
Response: Thank you for your positive feedback on the selection of references. We are glad that the inclusion of global studies by WHO, as well as studies from Ethiopia and culturally similar African nations, was found to be appropriate and relevant. Additionally, we appreciate your acknowledgment of the references on qualitative research methods and thematic analysis.
Comment 7: Table 1 is merely sample description and table 2 lists the themes and sub-themes. Nothing further can be added to them.
Response: Thank you for your feedback. We agree that Table 1 and Table 2 are focused on providing essential information—the sample description and the identified themes and sub-themes, respectively. We have corrected in the revised manuscript as follows:
Table 1: Characteristics of study participants
Tables 2: Themes and Sub-themes
In sum, I found this to be a well-designed study and well-written manuscript.
Response: Thank you for your appreciation.
Reviewer 2 Report
Comments and Suggestions for Authors
Well done on producing a high-quality and well-written article on an important topic. I have a few suggestions to ensure that the work will make an important contribution to the body of literature.
1) In the introduction please provide a clear definition of IPV that the paper operationalises. The introduction is engaging and concise, and this definition will assist readers in understanding IPV and the findings of the study.
2) The paper states that it wishes to engage with the cultural norms that underpin IPV, which is an important feature of IPV. However, it only lightly outlines these cultural norms. It would be good to see these engaged with more thoroughly with named examples.
3) Great to see the theoretical frame outlined, could a sentence be added that outlines why this theoretical frame is suitable for the topic? The methods are very comprehensively outlined and clearly written.
4) The results are well-presented, however could be improved by assessing how the positions of the different groups may have impacted the findings. For example, are the male focus group participants perspectives coming from a different type of knowledge to the practitioner perspectives? And while all these types of knowledge are valid - when there are differences how are they navigated in your study?
5) Please consider revising the heading 'causes' of IPV, to better fit your argument (for example is it the alcohol causing the violence or is the cultural factor a cause and alcohol an exacerbating factor).
Thank-you for your hard work on this important topic.
Author Response
Dec 24, 2024
For review article: Community Perspectives on Intimate Partner Violence During Pregnancy: A Qualitative Study from Rural Ethiopia
Dear reviewer,
Dear reviewer, we would like to thank for your investing time and efforts made to provide the valuable and very crucial feedback to improve our manuscript. We would like to acknowledge that all the comments and questions raised are really important to further improve our manuscript. We have carefully addressed all the comments and questions raised point by point. We highlighted with a red color in the revised manuscript.
Response to Reviewer #2
Well done on producing a high-quality and well-written article on an important topic. I have a few suggestions to ensure that the work will make an important contribution to the body of literature.
Response: Thank you for your appreciation.
Comment 1: in the introduction please provide a clear definition of IPV that the paper operationalizes. The introduction is engaging and concise, and this definition will assist readers in understanding IPV and the findings of the study.
Response: Thank you for your insightful feedback. We have defined IPV that the paper operationalizes in the introduction section of the revised manuscript as follows:
Intimate partner violence refers to any behavior within an intimate relationship that causes physical, psychological, or sexual harm including acts of physical aggression, sexual coercion, psychological abuse, and controlling behaviors, typically by a current or former intimate partner or spouse [1].
Comment 2: The paper states that it wishes to engage with the cultural norms that underpin IPV, which is an important feature of IPV. However, it only lightly outlines these cultural norms. It would be good to see these engaged with more thoroughly with named examples.
Response: Thank you for your insightful feedback. We appreciate your observation regarding the engagement with cultural norms underpinning IPV. We have included examples of cultural norms, beliefs and the societal expectations that contribute to the occurrence of IPV in the revised manuscript as follows:
An important factor in the prevalence of IPV is cultural norms, such as the traditional beliefs that men are superior to women and that a husband has the right to discipline his wife [16, 17]. Beliefs that IPV is a private and family matter and societal attitudes, such as accepting IPV and the normalization and stigma surrounding reporting IPV, also play a pivotal role in its occurrence [18-20]
Comment 3: Great to see the theoretical frame outlined, could a sentence be added that outlines why this theoretical frame is suitable for the topic? The methods are very comprehensively outlined and clearly written.
Response: Thank you for your appreciation and insightful feedback concerning the theoretical framework. We have added the sentence that outlines why the socio-ecological framework is suitable for exploring the community perspectives towards IPV during pregnancy in the revised manuscript as follows:
The choice of this framework is rooted in its ability to holistically capture the complex interplay of factors influencing IPV, particularly in rural settings where cultural norms and community structures are pivotal [24].
Comment 4: The results are well-presented, however could be improved by assessing how the positions of the different groups may have impacted the findings. For example, are the male focus group participants’ perspectives coming from a different type of knowledge to the practitioner perspectives? And while all these types of knowledge are valid - when there are differences how are they navigated in your study?
Response: Thank you for your appreciation. We would like to clarify that practitioners were not included in this study. Our participant groups comprised male partners, community leaders, women's development army members, and pregnant women who experienced IPV. Really, assessing the perspectives of IPV from multiple perspectives (groups) has positively impacted our findings.
Comment 5: Please consider revising the heading 'causes' of IPV, to better fit your argument (for example is it the alcohol causing the violence or is the cultural factor a cause and alcohol an exacerbating factor).
Response: Thank you for pointing this out. We agree with this comment. Therefore, we have revised causes of IPV by contributing factors of IPV during pregnancy.
Thank-you for your hard work on this important topic.
Response: Thank you for your appreciation.